# Visual stimulation with food pictures in the regulation of hunger hormones and nutrient deposition, a potential contributor to the obesity crisis

**Kalina Duszka**[1]*, **András Gregor**[1], **Martin Willibald Reichel**[1], **Andreas Baierl**[2], **Christine Fahrngruber**[1], **Jürgen König**[1]

**1** Department of Nutritional Sciences, University of Vienna, Vienna, Austria, **2** Department of Statistics and Operations Research, University of Vienna, Vienna, Austria

* kalina.duszka@univie.ac.at

**Data Availability Statement:** All relevant data are within the paper and its Supporting Information files.

## Abstract

Food cues affect hunger and nutritional choices. Omnipresent stimulation with palatable food contributes to the epidemics of obesity. The objective of the study was to investigate the impact of food cues on appetite-related hormones and to assess the functionality of the secreted hormones on macronutrient uptake in healthy subjects. Additionally, we aimed at verifying differences in the response of total and active ghrelin to stimulation with food pictures and to a meal followed by the stimulation. We were also interested in the identification of factors contributing to response to food cues. We recruited healthy, non-obese participants for two independent cross-over studies. During the first study, the subjects were presented random non-food pictures on the first day and pictures of foods on the second day of the study. Throughout the second study, following the picture session, the participants were additionally asked to drink a milkshake. Concentrations of blood glucose, triglycerides and hunger-related hormones were measured. The results showed that concentrations of several hormones measured in the blood are interdependent. In the case of ghrelin and gastric inhibitory peptide (GIP) as well as ghrelin and glucagon-like peptide-1 (GLP-1), this co-occurrence relies on the visual cues. Regulation of total ghrelin concentration following food stimulation is highly individual and responders showed upregulated total ghrelin, while the concentration of active ghrelin decreases following a meal. Protein content and colour intensity of food pictures reversely correlated with participants' rating of the pictures. We conclude that observation of food pictures influences the concentration of several appetite-related hormones. The close link of visual clues to physiological responses is likely of clinical relevance. Additionally, the protein content of displayed foods and green colour intensity in pictures may serve as a predictor of subjective attractiveness of the presented meal.

**Funding:** AG H-211758/2018, Hochschuljubiläumsstiftung der Stadt Wien, https://www.wien.gv.at/recht/gemeinderecht-wien/fonds-stiftungen/stiftungen/wissenschaft.html The funders had no role in study design, data collection and analysis, decision to publish, or preparation of the manuscript.

**Competing interests:** The authors have declared that no competing interests exist.

## Introduction

Most food consumption in western populations happens for reasons other than energy shortage, suggesting that a significant proportion of food consumption is driven rather by pleasure than by a physiological need [1]. Dietary behaviours and the drive to eat are undeniably powered by food-related cues. Food cue stimulation is omnipresent throughout the day. Advertisements, an abundance of products in appetising packages in stores but also the sight or smell of food, people eating or talking about food as well as emotions, feelings and activities can represent cues. The number as well as the density of the food-related cues have been implicated in choices to consume food or restrain from it [2–5]. It has been suggested that exposure to cues triggers the expectation of rewards in the form of food, which might be misinterpreted as hunger [6]. Importantly, food choices consumed as snacks besides main meals appeared to be more influenced by food-related cues than main meals and that internal and external cues, rather than hunger, are the most frequently recalled triggers of snacking behaviour. Additionally, snacks are higher in energy and lower in nutrient content than regular meals eaten during meal times [7–9]. A response to a food-related stimulus seems to be linked to higher overall caloric intake and, accordingly, to the development and maintenance of obesity and obesity-related health problems [5, 9]. Obesity is a major contributor to the development of diseases relevant to modern society including metabolic syndrome, cardiovascular disease, cancer, autoimmune diseases, dementia, depression and many others [10–15]. Therefore, understanding regulations that lead to weight gain on the level of unconscious processes as well as molecular mechanisms is of high importance.

The response of the gastrointestinal tract to the sight, smell, taste, or even thought of food is referred to as cephalic phase responses (CPRs). CPRs are preabsorptive, innate and learned physiological reactions that prepare the gastrointestinal tract for the optimal processing of ingested foods. Additionally, CPRs are associated with increased salivation, secretion of digestive enzymes, gastric acid, gastrin and insulin [16]. Therefore, stimulation with food cues results in increased levels of diet-related hormones in blood [17–20], however, the functionality of these triggered hormones and their contribution to nutrient uptake has never been assayed.

The major hunger-signalling hormone ghrelin regulates dopamine levels and neuron activity in the brain enhancing food-seeking behaviours, mediating rewarding properties of food, increasing the appeal of high-caloric foods [21–25] but also prompting other self-indulgent traits including gambling [26]. Similarly, alcohol consumption [27, 28], acute smoking [29, 30] and drug use [31, 32] have been shown to increase ghrelin levels. Ghrelin is referred to as hunger hormone, however, it is more likely an anticipation hormone which concentration changes according to the expected meal schedule or the presentation of food leading to meal initiation [33–37]. This mechanism of ghrelin concentration regulation is impaired in obese, anorexic and bulimic individuals [38–41]. Moreover, obese individuals are characterised by decreased levels of circulating ghrelin [42, 43] which most likely serves as a signal of nutrients excess. However, HFD-triggered obesity in mice is accompanied by ghrelin resistance [44] which may contribute to dysregulation of hunger and satiety perception. Ghrelin is secreted into the bloodstream in response to food cues [17, 19] and drastically drops 15–30 min after sham feeding [45]. However, it is not known whether ghrelin is secreted in its acylated or dis-acetylated form in response to stimulation with food. Initially, the acylated version of ghrelin was considered biologically active. However, recently more evidence indicates that signaling of both desacylated and acylated ghrelin results in distinct as well as complementary physiological outcomes [46–49]. Nevertheless, the exact roles and signaling pathways of each version of ghrelin, particularly in the context of hunger signaling, remain to be determined.

Incretins, glucose-dependent insulinotropic peptide (GIP) and glucagon-like peptide-1 (GLP-1) stimulate insulin release in response to a meal and all these factors together coordinate blood glucose levels [50, 51]. GIP is synthesized and released from the enteroendocrine K cells in the proximal intestine in response to glucose but also to fat [52]. Consequently, it functions as an incretin, thus, it affects glucose uptake but also plays a role in lipid metabolism. The impact of food cues on GIP secretion has never been reported and its role in hunger perception remains controversial [53, 54]. GLP-1 has insulinotropic as well as glucagonostatic activities, it inhibits gastric emptying and decreases food intake [55]. Postprandial release of GLP-1 results in the reduction of appetite and the inhibition of food intake [56]. It has been proposed that endogenous production of GLP-1 alone could decrease the food cue-induced activity of a major reward system, the orbitofrontal cortex, resulting in satiety [57]. Finally, insulin, the main regulator of blood glucose levels, also regulates satiety and food-seeking behaviours [58]. However, infusion of insulin does not affect hunger and food cravings already triggered by exposure to food pictures [59]. The release of insulin in response to view and thought of food has been described many years ago [58]. Insulin modulates signalling in dopaminergic reward circuits in the brain and insulin resistance has been associated with diminished levels of dopamine in different parts of the brain [60–65]. Since dopamine is an important modulator of food as well as drug reward and consumption [66], the interaction between insulin and dopamine was suggested to underlie "food addiction" [64]. Moreover, insulin itself has been shown to have rewarding properties [67, 68].

The aim of this study was to investigate the impact of food cues on the secretion of appetite-related peptides. Further, we wanted to assess the functionality of these peptides in basic conditions as well as upon meal consumption by assessing blood glucose and triglycerides levels. Thus, we merge previously reported approaches analysing reaction to food cues or postprandial responses. Additionally, we sought to find what form of ghrelin, acylated or desacylated is secreted in response to food pictures in order to contribute to unveiling the roles of the different forms of ghrelin in hunger signaling. Finally, we sought out to identify which visual factors in the food cues stimulation contribute to triggering the response of the viewers.

## Materials and methods

### Study I

The participants of the study were recruited by posters and leaflets distributed at the University of Vienna. Also, the information about the study was published on social media pages of the students of the university. In study I 25 participants registered and 23 finished the study (S1 Fig), of which 3 were male and 20 were female aged 19–28 years (mean±standard deviation (SD) = 23.43±2.9) (S1 Table). The participants' body weight was 48–78 kg (61.62±7.8) and their BMI 19.36–25.53 kg/m$^2$ (21.94±1.9). The female participants were questioned in a written form about possible pregnancy and were asked to report the day of their menstrual cycle to confirm the absence of pregnancy. However, it was not possible to coordinate the participants according to their cycle for the experimental days. The participants were included in the study based on their declarations of being overall healthy without known food allergies, digestive complains, metabolic or gastrointestinal diseases. Subjects suffering from any of the listed ailments were excluded from the study. The participants were required to follow a standard diet with milk products and/or meat. All the health and diet-related information was declared in a written form. The participants were informed that the study assesses "changes in the blood levels of intestine-derived hormones" along the day, to justify blood sampling and to partly inform about the study aim. The subjects were selectively not informed that the study involves analysis of the impact of the visual stimulation to avid the possible impact of conscious

processes on the spontaneous reaction to food cues. Instead, the participants were informed that they can request full information on the study design after the study is completed. The subjects filled and signed a written consent form confirming willingness to participate in the study with the possibility to withdraw anytime and agreeing to donate blood samples. The participants were invited two times for a one-day study with a minimum of five days between the two study days. The study followed a crossover design as each participant was submitted to control as well as stimulatory conditions. All participants were asked to consume meals regularly for seven days preceding the study and between the two study days: breakfast between 7:30 and 8:30 am, lunch between 12:00 and 1:00 pm. Subjects were asked to consume a light supper on the day preceding the studies and restrain from eating from 9:00 pm until 8:00 am the next day. They were also asked not to be sleep deprived to ensure good quality tests results. The protocol of the study is depicted in Fig 1A. At 8:00–8:30 am on the day of the study, a peripheral venous catheter was installed by specialized staff and blood samples were withdrawn. Each blood sample drawn during the study day was approximately 4 ml. Afterwards, the subjects were asked to fill out the hunger score form (very full-very hungry, scale 1–5) and breakfast was provided (two slices of cheese, tomato, jam, two slices of bread and tea; approximately 650 kcal). For the next 2 h, the participants were asked to restrain from food but were allowed to drink 0.5 l water. At 10:50 am the subjects were asked to fill out the hunger score form again and blood samples were drawn. At 11:00 am the participants took a 15 minutes computer test. The computer test with pictures and questions was designed and performed using FIZZ software (Biosystem, France). On the first day of the study, they were submitted to a control protocol with pictures of random non-food objects (description of the pictures is made available in S2 Table). The pictures were changing every 15 seconds and the participants were asked to rate each picture using an in-house designed questionnaire with a scale of 1–5 on attractiveness and arbitrary liking ("Do you find the picture pleasant?" (unpleasant-pleasant, scale 1–5), "How do you feel when looking at the picture?" (unwell-well, scale 1–5), "Do you think that this object smells good?" (no-yes, scale 1–5)). The questions rating the pictures were displayed directly under each picture. During the second day of the study, the participants were shown pictures of foods. The pictures contained processed as well as unprocessed foods, of low or high caloric content, and ranged from vegetables and fruits, through meats, snacks and desserts to fast foods (description of the pictures is made available in S3 Table). All of the pictures used were obtained from the Food-pics image database [69]. The study subjects were asked to answer questions rating from 1–5 the attractiveness, possible taste and smell of the presented food ("How do you feel when looking at the picture?" (unwell-well, scale 1–5), "Do you think that this object is tasty?" (no-yes, scale 1–5), "Do you think that this object smells good?" (no-yes, scale 1–5)). Each picture was presented for 15 seconds. Afterwards, blood samples were drawn at 11:15, 11:30, 11:45 am, 12:00, 12:15 and 12:30 pm with the aim of measuring levels of glucose, triglycerides and hunger-related hormones.

## Study II

For study II a group of 23 subjects, separate from participants of the study I were recruited. Out of them, 20 completed the study with one male and 19 females aged 18–28 years (23.05 ±3.1; S1 Table). The participants' body weight was 45.5–74 kg (59.5±8.2) and their BMI 16.89–26.95 kg/m$^2$ (21.3±2.6). Study II followed the exact design of the first study with the difference that after the neutral (first day of the study) or food (second day of the study) picture sessions the participants were given a milkshake (Fig 1B). The subjects were informed that they would be given breakfast and a portion of milkshake so that each day of the study, including controls they expected both meals. Additionally, the participants were informed that the study

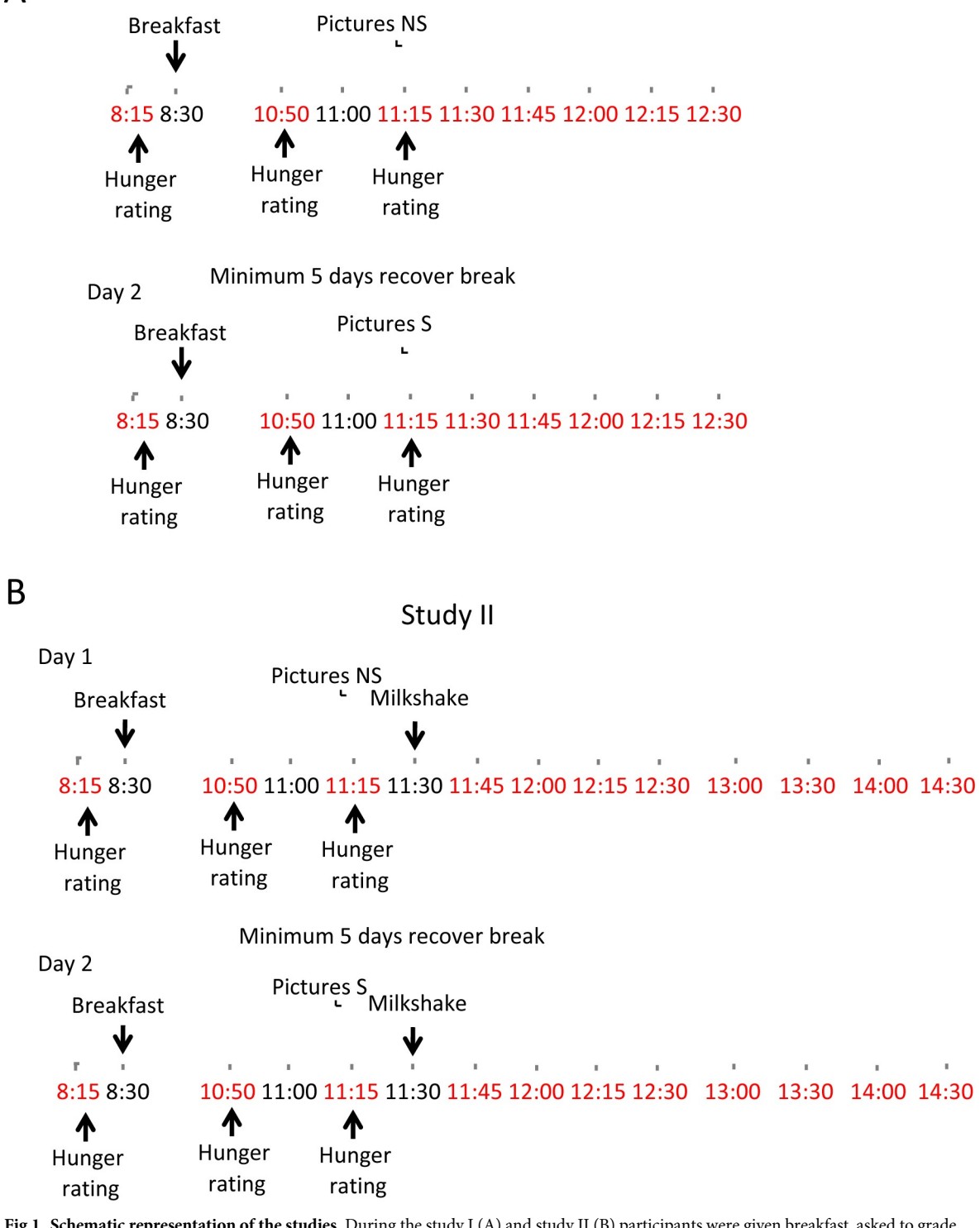

**Fig 1. Schematic representation of the studies.** During the study I (A) and study II (B) participants were given breakfast, asked to grade their hunger and to rate pictures of neutral objects or of food. For study II, following the picture session, the participants were given milkshake. The time points marked red indicate blood collection.

investigates the impact of the nutrients in the milkshake on the hormone levels in the blood. Similarly like in study I, the subjects were not informed that the impact of the visual stimulation would be analysed. The milkshake was given at 11:30 and the participants were asked to consume it within 3–5 minutes. The milkshake was made in-house by blending milk, cream and vanilla ice cream, all lactose-free. Its nutritional value was approximately 765 kcal with 25 g of fat, 120 g of carbohydrates and 15 g of protein. The total volume of the milkshake was around 400ml. Blood samples were drawn at time points 15min (11:45), 30min (12:00), 45 min (12:15), 60 min (12:30), 90 min (13:00), 120 min (13:30), 150 min (14:00), 180 min (14:30) after consumption of the milkshake. Similarly as study I, the participants were asked to rate the viewed pictures and report hunger.

The presented experimental design of human studies, including delivering partial information to the participants concerning the study aim, has been approved by the ethics committee of the University of Vienna (Ethikkommission, Besondere Einrichtung für Qualitätssicherung, Universität Wien).

## Samples analysis

Fresh human blood samples were used to measure glucose (Accu-Chek Performa, Roche, Mannheim, Germany) and triglycerides (Accutrend Plus and Accutrend Triglycerides, Roche). The remaining blood samples were collected into low-binding tubes and mixed with EDTA, aprotinin (0,6 TIU/ml of blood, Sigma Aldrich, St. Louis, MO, USA) and DPPIV inhibitor (Merck, Darmstadt, Germany). Another tube additionally containing HCl (0.05 N final concentration) was used to collect blood for ghrelin measurements. All samples were immediately centrifuged at 4˚C, 3600 g for 10 min. Plasma was collected, frozen and stored at -80˚C. Samples from selected time points were used for quantification of total ghrelin (ELISA), acylated ghrelin (ELISA), insulin, Peptide YY (PYY), GIP, GLP-1 and glucagon (Milliplex, all from Merck).

## Statistics

Statistical analysis for hunger ratings and plasma hormone concentrations was performed using SPSS 23.0 (IBM, NY, USA). For data sets with multiple timepoints repeated measures ANOVA with Bonferroni correction for multiple testing was used to assess the effect of type of pictures and time. For data sets without multiple time points, comparison of two groups were performed applying two-sided student's t-tests. For all data sets 95%-confidence intervals and Cohen's d effect estimates were calculated. Linear regression analysis was performed to determine correlations between hunger score, plasma metabolites, plasma hormones and the computer test results. *P* values equal 0.05 or lower were considered to be significant.

The raw data supporting the conclusions of this manuscript will be made available by the authors, without undue reservation, to any qualified researcher.

## Results

### Stimulation with food pictures affects hunger perception, blood glucose and hormones concentrations

The study participants were invited on two days to record their response to pictures of neutral objects (not stimulating, NS, S2 Table) or pictures of appetising food (stimulating, S, S3 Table). The subjects rated the questions concerning the food images higher (p<0.001; Cohen's d = 0,43; 95% CI: 0.67, 1.54) than the neutral figures photographs (Fig 2A). 2-way mixed ANOVA with within-subject factor "hunger rating" and between-subject factor "pictures" using Huynh-Feldt correction of sphericity was applied. There was a significant main effect for

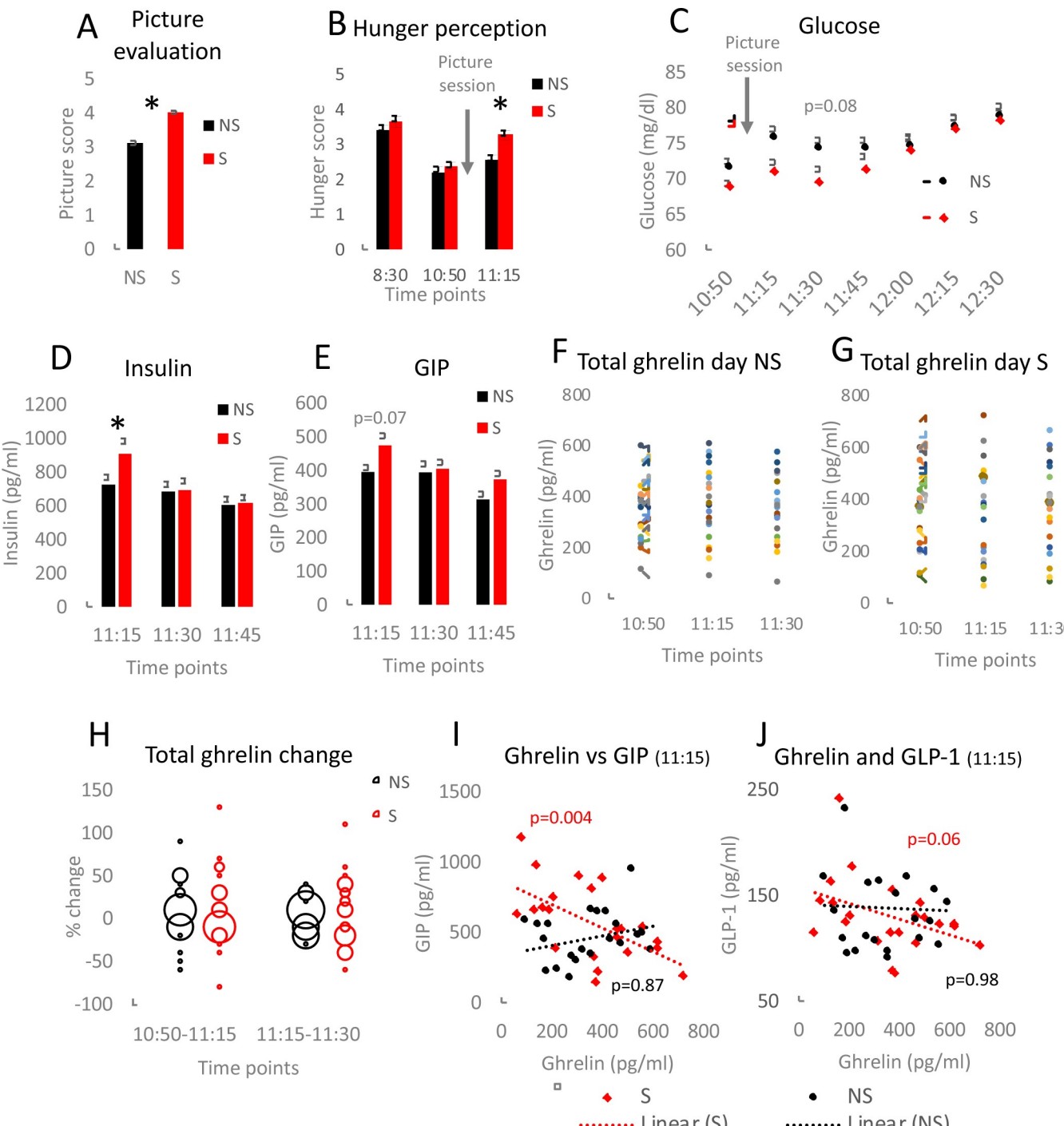

**Fig 2. Food cues affect blood glucose and appetite-replated peptides.** The participants rated different aspects of the pictures on a scale 1–5 (A). Hunger level was reported on a scale 1–5 by the participants at the indicated time points (B). Glucose levels were measured at all blood sampling time points (C). Insulin (D) and glucose-dependent insulinotropic peptide (GIP) (E) concentrations were quantified in blood collected at the indicated time points. For panel A statistical significance was analysed using two-tailed Student's t-tests. Data (A-E) are presented as the mean±SEM; *p<0.05. For panels D-E ANOVA with Bonferroni correction for multiple testing was applied. Total ghrelin levels were measured in the samples collected on NS (F) and S (G) day of the study. Each line represents one participant. Distribution of total ghrelin concentration changes was evaluated by comparing % difference between two indicated time points (H). Linear regression analysis was performed to verify the correlation between ghrelin and GIP (I) as well as ghrelin and glucagon-like peptide-1 (GLP-1) (J) for blood samples collected at 11:15; n = 23.

hunger perception (F(1.84; 81.11) = 62.78, p<0.001) and a significant hunger*pictures interaction (F(1.84; 81.11) = 3.43, p<0.05. Further analysis indicated that exposure to the pictures of appetising food compared to watching the neutral pictures resulted in an increased subjective hunger perception (time point 11:15; p = 0.0006; Cohen's d = 1.12; 95% CI: 0.31, 1.08; Fig 2B). In order to assess the impact of food cues, the concentrations of glucose, triglycerides and appetite-related factors were measured in the participants' blood. There was no statistically significant difference in blood glucose between the NS and S day (Fig 2C). However, the participants on the S day tended to have lower glucose levels than NS after viewing the picture (11:30 am time point, p = 0.08; d = 0.68; 95% CI: -0.64, 10.22). For assessing insulin and GIP development, 2x3 repeated measures ANOVA with Greenhouse-Geisser correction were used. There was a significant main effect for insulin (F(1.45; 62.42) = 10.65, p<0.001) but no significant insulin*picture interaction. For GIP there was a significant GIP*picture interaction (F(1.38; 26.29) = 10.06, p = 0,002) as well as significant within-subject changes of GIP concentration from the time point 11:15 to 11:30 (F(1;19) = 12.71, p = 0.002) and the time point 11:30 to 11:45 (F(1;19) = 10.30, p = 0.005). Food stimuli compared to the non-food objects resulted in statistically non-significant trends indicating increase in blood insulin (p = 0.05; d = 0.9; 95% CI: 4, 430; Fig 2D and S4 Table) and GIP level (p = 0.07; d = 0.8; 95% CI: 39.7, 121.5; Fig 2E and S4 Table). These trends were observed directly after the picture exposure (time point 11:15) and were not present at further time points following the stimulation. Blood concentrations of triglycerides, GLP-1, PYY, glucagon and total as well as active ghrelin did not differ between the two days of the study (S2A–S2C Fig and S4 Table). However, a linear correlation was measured between GLP-1 and glucagon as well as PYY and glucagon across all measured time points (11:15 am, 11:30 am, 11:45 am), which was independent from the type of pictures shown (S2D–S2I Fig). Importantly, despite the lack of a significant change between average ghrelin concentrations, there was high variability between the subjects' ghrelin concentrations (Fig 2F–2G). Compared to the NS day of the study, results from the S day showed more changes, more increase as well as more decrease in ghrelin blood concentrations from the time point before picture session to directly after the session (10:50 am-11:15 am) and within half an hour after the session (11:15 am-11:45 am; Fig 2F–2H). The differences in the distribution of concentration changes between the NS and the S day were much less pronounced for active ghrelin (S2J Fig). Interestingly, total ghrelin was reversely correlated with GIP and GLP-1 in samples collected on the day S and this correlation was absent in NS samples (Fig 2I and 2J).

## Stimulation with food pictures affects glucose disposal and incretins response

In the second study, we aimed at verification of the functionality of the hormones secreted during sensory stimulation. Similar to study I, the food pictures obtained higher test score compared to the neutral pictures (p<0,001; d = 1,16; 95% CI: 0.89, 0.92; Fig 3A). Likewise, similarly to study I, food cues compared to the pictures of neutral objects increased hunger perception (Fig 3B). However, this time the difference was not statistically significant (p = 0.06; d = 1,1; 95% CI: 0.51, 0.88). In order to verify a functional impact of the sensory stimulation on blood glucose and hormones, the participants were given a milkshake succeeding exposure to the neutral pictures or the food pictures. The liquid form of the meal allowed fast consumption and the rapid intake of macronutrients resulted in an increase of both glucose and triglycerides in the blood (Figs 3C and S3A). The concentration of triglycerides was comparable between the two days of the study (S3A Fig). The increase in glucose level was more pronounced on NS compared to the S day of the study with a significant difference (p = 0.03, d = 0.7; 95% CI: 1.37, 16.7) an hour after the milkshake intake. The blood glucose

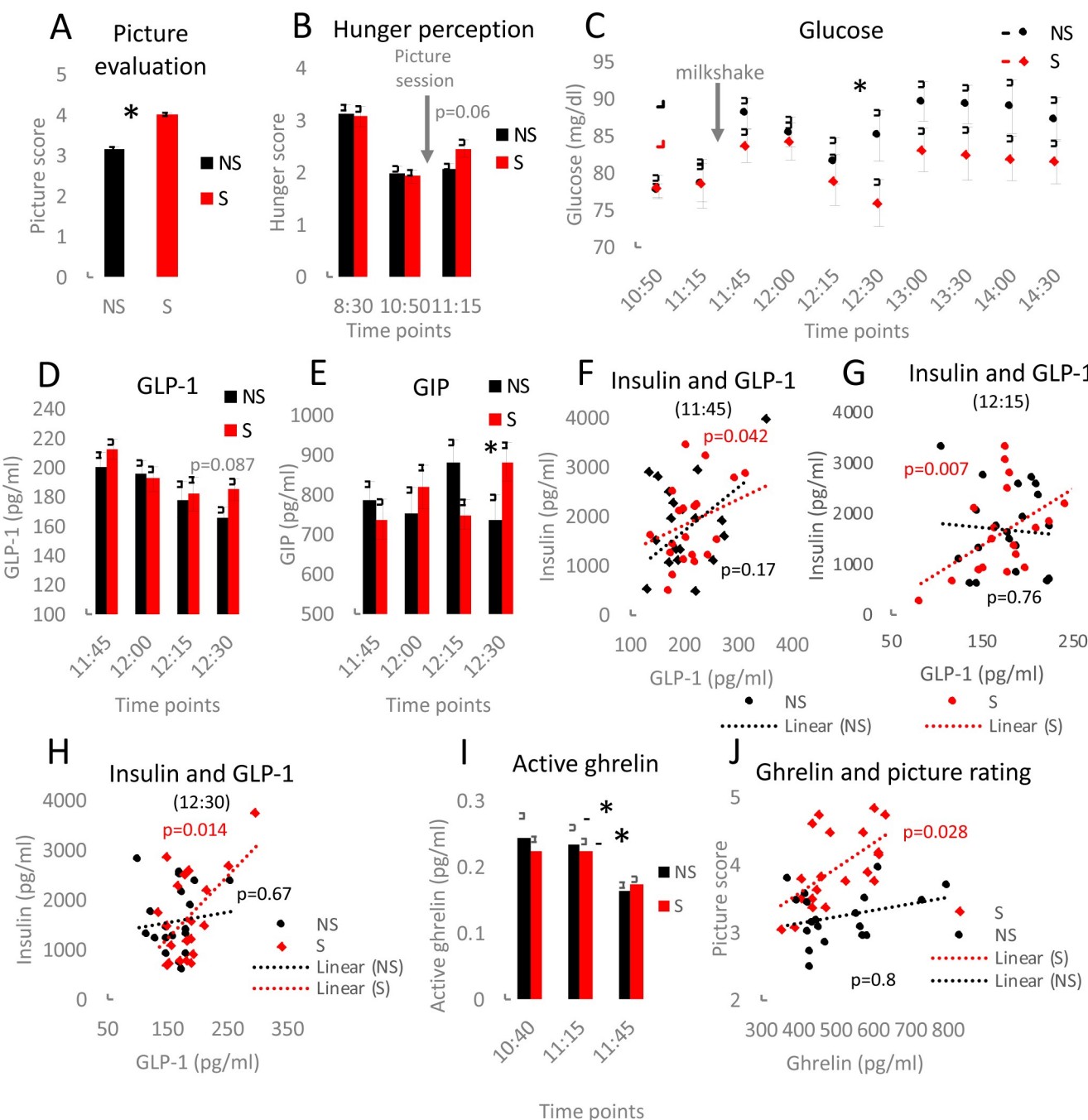

**Fig 3. Exposure to food pictures influences postprandial blood glucose and appetite-related peptides.** The participants ranked the pictures on a scale 1–5 (A). Hunger level on a scale 1–5 was rated by the participants at the indicated time points (B). Glucose levels were measured at the specified sampling time points (C). GIP (D) and GLP-1 (E) concentrations were quantified in blood samples collected at the indicated time points. Linear regression was used to validate the relationship between insulin and GLP-1 (F-H) for blood samples collected at the chosen time points. The concentration of active ghrelin in blood samples was assessed (I). Linear regression was applied to analyse interconnection between total ghrelin and picture score (J). Two-tailed Student's t-tests were used to compare the experimental groups of panel A; n = 20, *p<0.05. For panels B, D, E and I ANOVA with Bonferroni correction for multiple testing was applied. Error bars represent the mean±SEM.

reached the lowest concentration 45 minutes after the consumption of the milkshake on the NS day and with 15 minutes delay (60 minutes after the milkshake ingestion) on the S day.

Based on ANOVA analysis, there was a significant pictures*GIP interaction ($F_{(1.96; 37.18)}$ = 7.36, p = 0.002). However, post hoc analysis indicated a statistically non-significant trend towards increased concentrations of GIP (*p* = 0.05; d = 0.7; 95% CI: -2.6, 294.2) and GLP-1 (*p* = 0.087; d = 0.6; 95% CI: -3, 42.6) on the S compared to the NS day (Fig 3D and 3E and S5 Table). Interestingly, the differences in GLP-1 and GIP concentrations occured only as late as 1 hour after the milkshake consumption, at the same time point when the difference in glucose levels between NS and S was the most pronounced. Blood concentrations of insulin, glucagon and PYY were not affected by the type of presented pictures at the tested time points following the milkshake consumption (S5 Table). Postprandial insulin concentrations correlated positively with GIP (12:15 pm time point, S3B Fig) as well as with GLP-1 (11:45 am, 12:15 pm, 12:30 pm; Fig 3F–3H) and glucose (11:45 am, 12:00 pm, 12:15 pm, 12:30 pm; S3C–S3F Fig). Importantly, only food pictures triggered the correlation between insulin and GLP-1 or insulin and GIP. Milkshake consumption was followed by a decrease in blood concentrations of active ghrelin on both days of the study (significant effect of time ($F_{(1;16)}$ = 9.65, p = 0.007); Fig 3I) while total ghrelin remained unaffected (S4A Fig). Similar to study I, there were no significant differences in total and active ghrelin between the stimulation with neutral objects or food pictures. The variability in ghrelin concentration changes between 10:50 am and 11:15 am (S4B Fig) were comparable to those measured for study I. However, upon receiving the milkshake (11:15 am-11:45 am) the distribution of changes strongly diminished while the variability of the concentration of active ghrelin was not affected by the meal (S4C Fig). Total ghrelin in study I showed a significant positive correlation (*p* = 0.028) with the ranking of the food pictures (Fig 3J) directly after the exposure to the pictures (11:15 am). A similar trend, however, not significant (*p* = 0.086) was observed in study II (S4D Fig). While no correlation was observed between total ghrelin and the rating of neutral pictures.

In addition, the rating of the food pictures corresponded to hunger perception (S4E Fig). Surprisingly, in both experiments the positive evaluation of the food pictures correlated reversely with the protein content of the foods (study I *p* = 0.004, study II *p* = 0.002; Figs 4A and S4F). The participants scored pictures depicting desserts, sweets and fruits higher, which were particularly low in protein, whereas meats, fish and meat-based fast foods were rated relatively low (S2 Table). Moreover, low-calorie foods were rated higher than such with high-calorie content (Fig 4B). Other nutritional features like fat and carbohydrate content did not correlate with the pictures' scores (S4G and S5H Figs). In the case of the neutral objects, the intensity of colours present in the pictures did not influence the participants' choice. In contrast, colours in the food pictures affected the picture score. Namely, more green colour decreased preference towards the presented foods and lower apperance of green colour was favoured (Figs 4C and S4I). The intensity of green colour did not correspond to any nutritional information tested (protein, fat, carbohydrates, calorie content; S5A–A5D Fig).

## Discussion

Within our study, we investigated several aspects of the response of young, healthy subjects to visual food stimulation. In study I, average blood concentration of total ghrelin was not affected by the visual stimulation when considering the whole group. However, analysing the response of singular participants, we realized a high variability in response to food pictures which was not present for the neutral pictures. This observation indicates that the participants' response is very individual and also points out that some participants were non-responders which was characterised by the lack of a decrease or increase of ghrelin concentration. However, for some of the responder participants, the increase in ghrelin levels is very rapid, resulting in a prompt peak (15 min after the stimulation) followed by a decrease (30 min after the

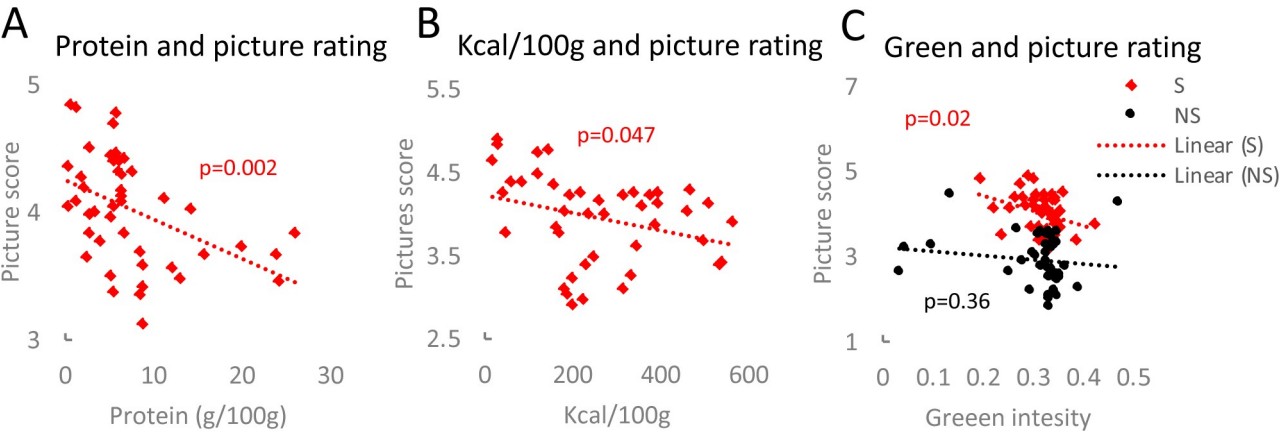

**Fig 4. Participants' preference towards pictures showing low protein, low calorie-dense foods with low green colour intensity.** Linear regression was applied to analyse interconnection between pictured food protein content and corresponding rating (A), caloric content per 100g of the shown food and picture score (B) as well as the intensity of green colour and picture evaluation (C).

stimulation) while in other subjects the concentration is rising with a delay. Therefore, the development of ghrelin levels over time is highly individualised. Overall, the elevation of total ghrelin concentrations in responders corresponds with previous reports [19]. Moreover, we demonstrate that the concentration of total ghrelin correlates with the score of food picture proving the impact of the pictures and revealing that in ghrelin non-responders reaction to food pictures was diminished.

Previously, it has been reported that ghrelin is released in response to food cues [19] however, it was not clear which form of ghrelin is concerned. We report that in contrast to active ghrelin, the concentration of total ghrelin was affected by the food cues. Total ghrelin concentrations are known to decrease after consumption of carbohydrate-rich food and to increase following protein consumption [70]. In contrast, acylated ghrelin levels drop after each type of meal, particularly after a protein- and fat-rich meal [71]. In study II, we decided to apply food composed of mixed macronutrients to study the response to an actual meal in a more representative way. We did not observe a decrease in total ghrelin levels 15 minutes after the meal. According to a previous study, which applied a meal comparable to ours [72], we could expect total ghrelin to decrease at the later time points. Contrary to total ghrelin, active ghrelin concentration decreased in response to meal intake, in accordance with previous reports [73–75]. However, active ghrelin was not affected by food pictures. Thus, we describe a discrepancy between the response of total and acylated ghrelin with one being stronger dependent on food cues (preprandial) and the other on consumption (postprandial).

We also demonstrate that glucagon and GLP-1, as well as glucagon and PYY reversely correlate at both stimulatory and neutral conditions. However, only food cues trigger a negative correlation between ghrelin and GIP as well as ghrelin and GLP-1. Accordingly, a previous study reported that GIP-prompted a decrease in ghrelin release in rats [76]. Furthermore, only a meal in combination with stimulatory pictures elicited a positive correlation between insulin and GIP as well as insulin and GLP-1. Thus, the co-regulation operates differently in the presence and in the absence of an actual meal. Contrary to a previous report [19], we observed a near significant difference in insulin concentration following visual stimulation. Our observation of food picture-stimulated correlation between insulin and GIP as well as insulin and GLP-1 further suggests that food pictures may in fact affect insulin.

The correlation between glucagon and GLP-1 reported here is surprising due to the glucagonostatic properties of GLP-1 [77]. However, it is important to note that the correlation is

not influenced by the type of pictures and that neither concentrations of GLP-1 nor glucagon separately are affected by the pictures at any of the tested time points. Therefore, these results indicate that at basic conditions, individuals with naturally occurring high GLP-1 concentrations also show high glucagon concentrations and subjects with low GLP-1 concentration have correspondingly low glucagon concentrations. The nature of this co-occurrence is not clear yet and remains to be clarified.

GIP, GLP-1 and insulin are released in response to a meal and together coordinate blood glucose levels [50, 51]. Hence, the positive correlation between these factors was anticipated. We, as first, describe changes in the concentrations of GIP upon food picture stimulation. The consequences of the upregulation of insulin, GIP and GLP-1 in response to visual food cues are well reflected in terms of the differences in blood glucose levels. Particularly in study I, in which we observe a trend in glucose levels indicating the difference between the S and NS day 15 minutes after the exposure to pictures (11:30 am), confirming previous report [78]. Therefore, we propose a model in which food cues provoke GIP and GLP-1 secretion. This in return, prompts insulin release leading to lower blood glucose concentrations. Decreased glucose is signalized to the brain and results in increased hunger perception (Fig 5).

Several factors influenced participants' preference in the food picture computer test. The participants of both studies clearly preferred pictures of foods low in protein compared to foods high in protein. The participants rated items like sweets, desserts and fruits higher compared to meat-containing pictures. Other liking criteria were calorie content and, surprisingly, the intensity of green colour. Contrary to a previous publication [79], in our study higher caloric content of the food pictures decreased the attractiveness of the picture. However, in this case, palatability of the foods shown during the study may strongly influence the results. Colour of food reportedly influences taste perception and food consumption [80–83]. Altered food colour changes preference and affects perceived palatability [84]. Green is generally associated with healthy foods and natural products [85]. However, in our study pictures with higher green intensity did not necessarily show green vegetables (only 1 out of 45 pictures). The food composition of the items with low or high green intensity seems to be random. Therefore, indeed it may just be the colour intensity that influenced the participants' choices.

In modern society, commonly occurring consumption in front of tv and computer screens results in increased calorie intake and adiposity [86, 87]. Therefore, it is particularly important to underline great health benefits connected with focusing on a meal without disruption with external stimuli. Based on studies concerning mindful eating [88, 89] as well as on the here presented impact of the food cues on the efficiency of nutrients uptake, concentrating on food should be advised as a proper way of preparing for meal consumption. It would be of particular interest for further research to verify if similar to here described hormonal responses to food cues are functional in obese and diabetic individuals. In this context, the regulation of insulin release by pre-meal visual or olfactory cues could contribute to improved meal tolerance and therefore be of great clinical relevance. Moreover, exploration of different types of food cues (visual, olfactory, multiple senses) and its impact on various population groups (age, obesity, metabolic diseases, anorexia, etc.) could unveil additional regulatory mechanisms.

Besides important contribution, our study comes with a few shortcomings. Our participants' sample consists mostly of female volunteers. As genders differ in terms of appetite, taste preferences and food choices [90–92], it can be argued that the study reports the response to sensory stimulation from the female perspective. However, within our data sets there is no impact of gender on any of the investigated parameters. Therefore, the data likely represent both genders. Another drawback of the study is connected with the fact that taste perception, appetite and food cravings are affected by the menstrual cycle [93–97]. Unfortunately, for organizational reasons, we were unable coordinate the participation of the female volunteers

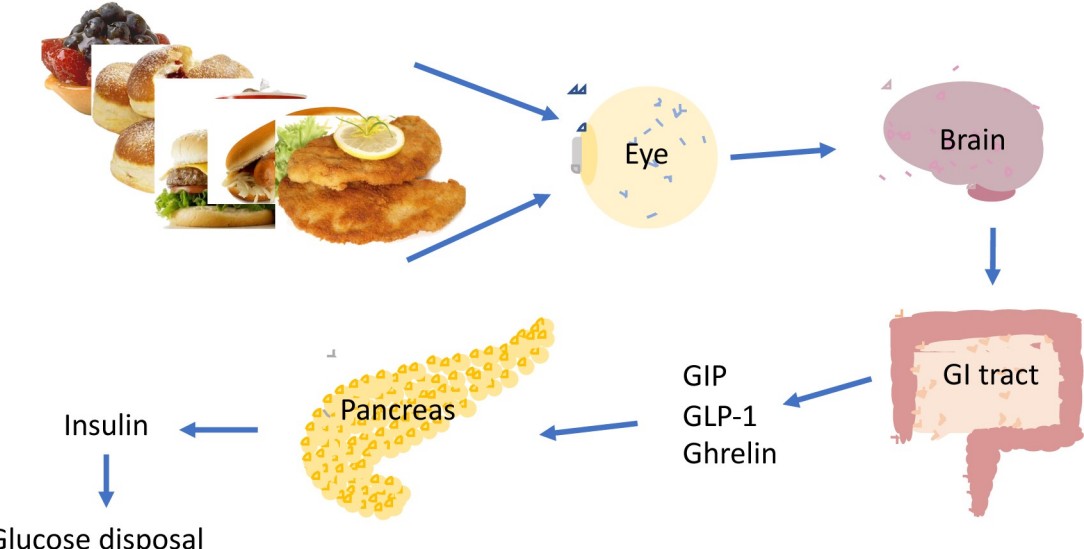

**Fig 5. Schematic representation of the hypothesis.** In study II, a small decrease in postprandial glucose concentration on S compared to NS day is observed for the first time point after the meal. Importantly, 1h after the meal, glucose concentrations drop further. This is most probably resulting from increased incretins and insulin concentrations and consequent glucose deposition. Interestingly, participants on the S day show delayed glucose decline and the most pronounced difference between NS and S day in glucose levels fits with the time point when the differences in GIP and GLP-1 concentrations between the two days of the study are the most pronounced. The differences between NS and S day imply that the efficacy of glucose uptake or deposition is affected by visual stimulation preceding a meal.

according to their menstrual cycle. Moreover, the group recruited for study II was characterized by a wide range of BMI (16–26 kg/m$^2$). Analysing the influence of BMI on response to visual stimulation, we found no impact for any of the examined parameters. We conclude that within the BMI range of our participants, there was no difference in the response to sensory stimulation in terms of hormone profiles, hunger ratings or pictures ratings.

Our study contributes to the unveiling of the impact of sensory stimulation on consumers' choices and the interconnection between various factors in hormonal hunger regulation. We as first, describe a study comparing the impact of food pictures on blood metabolites and hormones at basic conditions as well as consequences of visual stimulation on the metabolism of a meal. Although without strong statistical power, our results align with to-date studies showing an impact of visual cues on blood glucose and ghrelin concentrations. Additionally, we detected an interesting phenomenon concerning individual response to food pictures in terms of total ghrelin regulation. We report that total ghrelin is more affected by visual cues while acylated ghrelin is more affected by meal consumption. We also measured several correlations between hormones which were triggered by food pictures alone and others that needed both stimuli, the pictures and a meal. We also discovered that the protein content of foods is predictive for food preference and that the intensity of colours affects the attractiveness of food. Further studies are needed to unveil the mechanisms behind hormone interconnection and possible direct stimulatory effects. Additionally, verification of the functionality of here presented mechanisms in obese and diabetic individuals will be of high interest and clinical relevance.

## Supporting information

**S1 Table. Participants of study I and II.** Mean data are presented with±SD.
(DOCX)

**S2 Table. Pictures presented on day S of the study.** Food pictures presented a variety of products with indicated nutritional value and basic color intensity. The average participants' answers to the 3 questions evaluating the pictures are shown for the Study I and II.
(DOCX)

**S3 Table. Pictures presented on day NS of the study.** Pictures presented a variety of objects with indicated basic color intensity. The average participants' answers to the 3 questions evaluating the pictures are shown for the Study I and II.
(DOCX)

**S4 Table. Blood concentration of hunger-associated peptides in study I.** The data are presented with±SEM.
(DOCX)

**S5 Table. Blood concentration of hunger peptides in study II.** The data are shown as±SEM.
(DOCX)

**S1 Fig. Study flowchart.**
(PDF)

**S2 Fig. Exposure to pictures of food does not affect blood triglycerides, average total as well as active ghrelin.** Concentration of glucagon and GLP-1 as well as glucagon and PYY show correlation. Blood levels of triglycerides were measured between 10:50 and 12:00 on both days of the study (A). The concentration of total (B) and active (C) ghrelin were measured in the collected blood samples. ANOVA with Bonferroni correction for multiple testing was used to assess statistical differences. Data (A-C) are presented as the mean±SEM. Linear regression was analyzed to verify the connection between glucagon and GLP-1 (D-F) as well as glucagon and PYY (G-I) for samples collected at the indicated time points. Changes of the concentration of active ghrelin were calculated (J); n = 23.
(PDF)

**S3 Fig. Exposure to pictures of food does not affect postprandial blood triglycerides and correlation between insulin and glucose.** Blood levels of triglycerides were measured between 11:15 and 14:30 on both days of the study (A). Data are presented as the mean±SEM. Linear regression was analysed for correlation between insulin and GIP (B) as well as insulin and glucose for samples collected at 11:45 (C), 12:00 (D), 12:15 (E) and 12:30 (F); n = 20.
(PDF)

**S4 Fig. Exposure to food cues does not affect postprandial ghrelin.** Nutritional as well as visual features of the food pictures influence attractiveness of the food pictures. The concentration of total ghrelin was measured in the blood samples collected at the indicated time points (A). Distribution of the concentration of total (B) and active (C) ghrelin changes were evaluated by comparing % difference between two indicated time points. Linear regression analysis was performed to verify the correlation between total ghrelin and picture evaluation score (D). Linear regression was analyzed for the relationship between hunger rating and picture evaluation score (E), food protein content (F), fat content (E), carbohydrate content. (H) and picture evaluation score as well as green color intensity and picture evaluation score (I). ANOVA with Bonferroni correction for multiple testing was used to determine statistical significance for the data in panels A, E and F; n = 20, *p<0.05.
(PDF)

**S5 Fig. The macronutrient and energy content of presented foods does not correlate with the intensity of green colour.** Linear regression was analyzed for the relationship between

protein (A), fat (B), carbohydrate (C), caloric density (D) versus intensity of green colour. study II. The data are shown as±SEM.
(PDF)

## Acknowledgments

Authors would like to thank students for helping during the study: Kathrin Jurasek, Sarah Gringinger, Julia Kolonovits and Diana Melina Tanackovic.

## Author Contributions

**Conceptualization:** Kalina Duszka.

**Data curation:** Kalina Duszka, Martin Willibald Reichel, Andreas Baierl, Christine Fahrngruber.

**Formal analysis:** Kalina Duszka.

**Funding acquisition:** Kalina Duszka, András Gregor.

**Investigation:** Kalina Duszka, András Gregor.

**Methodology:** Kalina Duszka.

**Project administration:** Kalina Duszka.

**Resources:** Kalina Duszka.

**Software:** Martin Willibald Reichel.

**Supervision:** Kalina Duszka, Jürgen König.

**Validation:** Kalina Duszka, Andreas Baierl.

**Visualization:** Kalina Duszka.

**Writing – original draft:** Kalina Duszka.

**Writing – review & editing:** András Gregor, Martin Willibald Reichel, Andreas Baierl, Christine Fahrngruber, Jürgen König.

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
