## [Decision Letter · Decision Letter 0]

5 Dec 2019

PONE-D-19-23483

Visual stimulation with food pictures in the regulation of hunger hormones and nutrient deposition

PLOS ONE

Dear Dr. Duszka,

Thank you for submitting your manuscript to PLOS ONE. After careful consideration, we feel that it has merit but does not fully meet PLOS ONE’s publication criteria as it currently stands. Therefore, we invite you to submit a revised version of the manuscript that addresses the points raised during the review process.

We would appreciate receiving your revised manuscript by Jan 19 2020 11:59PM. To enhance the reproducibility of your results, we recommend that if applicable you deposit your laboratory protocols in protocols.io, where a protocol can be assigned its own identifier (DOI) such that it can be cited independently in the future. For instructions see: http://journals.plos.org/plosone/s/submission-guidelines#loc-laboratory-protocols

We look forward to receiving your revised manuscript.

Kind regards,

Jyu-Lin Chen

Academic Editor

PLOS ONE

Journal Requirements:

2. In your Ethics Statement, please provide:

a) Justification for not fully informing the participants about the purpose of the study

b) Clarification whether the IRB specifically approved not fully informing the participants

c) Clarification of how participants were debriefed at the end of the study."

3. In your Methods, please state the volume of the blood samples collected for use in your study.

4. In your Methods, please state where the participants were recruited for your study.

Additional Editor Comments (if provided):

Dear authors,

The manuscript has been reviewed by three reviewers. This scientific experiment has great potential to inform science. However, there are several areas needed to be clarify before this manuscript can be accepted for publication. The major concerns are related to methodology given there were two studies involved in this manuscript. Clear description of study design, study procedure, inclusion/exclusion criteria, and outcome variables are needed. In addition, in discussion section, the first paragraph is NOT supported by the study results. Addressing these issues can enhance the quality and clarity of the manuscript.

Reviewers' comments:

Reviewer's Responses to Questions

**Comments to the Author**

1. Is the manuscript technically sound, and do the data support the conclusions?

Reviewer #1: Yes

Reviewer #2: No

Reviewer #3: Yes

2. Has the statistical analysis been performed appropriately and rigorously? 

Reviewer #1: Yes

Reviewer #2: Yes

Reviewer #3: Yes

3. Have the authors made all data underlying the findings in their manuscript fully available?

Reviewer #1: Yes

Reviewer #2: Yes

Reviewer #3: Yes

4. Is the manuscript presented in an intelligible fashion and written in standard English?

Reviewer #1: Yes

Reviewer #2: No

Reviewer #3: Yes

5. Review Comments to the Author

Reviewer #1: This study has the potential to contribute significantly to the understanding of visual cues and hunger hormones. The outcome highlighting the highly individualized response to ghrelin has implications for further study and the understanding of hunger behavior in light of the obesity epidemic.

There are no major issues related to the literature review, methodology or presentation of the results. The discussion section could be expanded to include clinical implications of the outcomes and what additional research is needed to forward this science.

Minor issues are listed below.

Title:

This research appears an important contribution to the study of obesity, however, the title and key words do not highlight this point. I would think that the authors would like to increase the visibility among researchers looking at obesity.

Introduction:

Page 3, line 67: Suggest elaborating on negative health outcomes related to obesity that will support the significance of your research.

Page 4, line 77: Would remove the word hedonistic.

Page 4, lines 82-83: Would take the opportunity to discuss how ghrelin concentration is impaired in obese individuals.

Page 4, lines 84-86: Elaborate on the difference between active and inactive forms. What does the difference contribute to this research?

Page 5, lines 111-112: What will investigating what form of ghrelin add to your study?

Page 5, lines 117-123: How were the participants recruited? What were they told about the study? How did you ensure the study participants were not pregnant?

Methods:

Page 7. Lines 158-162: Recommend moving Figure 1 discussion to after 2.2 study II.

Page 7, lines 165-167: How were the participants recruited? Why did you recruit a difference sample size? How did you ensure the women were not pregnant?

Page 8, lines 184-185: How did you ensure validity and reliability of sampling and analysis procedures?

Results:

Page 14, lines 335-339: Figure 4 results do not appear to be included in the purpose and aims of the study.

Discussion:

Well developed. Would like to have seen a discussion on the clinical implications of the study’s outcomes.

Figure 1: Would include N and NS in labeling “pictures.”

Figure 2 F&G is confusing. State in key that each color is a participant.

Reviewer #2: Thanks for inviting me to review this article. It sounds an interesting topic but did not address the clinical implication of the study. The design seems not ethical for the participants, drawing eight times of blood sample without telling the purpose of the study. I did not find the statements of the ethical consideration. The writing style is not concise in methodology part. The gap of the literature was not addressed well.

Reviewer #3: An interesting study.

1. The description of the methodology can be further clarified. For instance, is this a two-arm experimental trial? or time-series with phase I and phase II? It is best to define what is the research design of this study.

2. There are also some unclear areas, such as in the abstract, the authors stated this is a two-day study. However, in the extensive description stated in the method, the days involved in the actual study spanned more than two days in study 1 and study 2. It would be helpful to have schematic diagram to display the start date and end date of the studies, and intervals between and within the studies.

3. Another unclear area: Are the participants in both study 1 and 2 the same group of sample, or they are pulled form different sampling groups?

Thanks.

6. PLOS authors have the option to publish the peer review history of their article (what does this mean?). If published, this will include your full peer review and any attached files.

Reviewer #1: No

Reviewer #2: No

Reviewer #3: Yes: Fang-yu Chou

---

## [Author Response · Author response to Decision Letter 0]

16 Dec 2019

First of all, we would like to thank the reviewers for positive comments and pointing out the weakness of the manuscript. We realized that the applied changes were indeed needed and that it significantly improved the manuscript. 

Journal Requirements:

The style requirements have been applied to the whole manuscript.

2. In your Ethics Statement, please provide:

a) Justification for not fully informing the participants about the purpose of the study

The participants were informed that the study assess “changes in the blood levels of intestine-derived hormones” along the day (study 1) or that the study investigates the impact of the nutrients in the milkshake on the hormone levels in blood (Study 2) to justify blood sampling and to partly inform about the study aim. The subjects were selectively not informed that the study involves analysis of the role of the visual stimulation to avoid the possible impact of conscious processes on the spontaneous reaction to food cues. Instead, the participants were notified that they can request full information on the study design after the study is completed. The subjects filled and signed a written consent form confirming willingness to participate in the study with possibility to withdraw anytime and agreeing to donate blood samples. The material and methods section was updated with this information.

b) Clarification whether the IRB specifically approved not fully informing the participants

The ethics committee specifically approved obscuring the full aim of the study. This information was inserted at the end of the description of the design of study 2.

c) Clarification of how participants were debriefed at the end of the study."

The participants were informed that if interested, they can receive full results of the measured parameters for their samples and that following the study completion they can receive a description of the study design including the full aim of the study.

3. In your Methods, please state the volume of the blood samples collected for use in your study.

Each blood sample drawn during the study day was approximately 4 ml. The materials and methods section was updated with the required information.

4. In your Methods, please state where the participants were recruited for your study.

The participants of the study were recruited by posters and leaflets distributed at the University of Vienna. Also, the information about the study was published on social media pages of the students of the university. The missing information was added in the materials and methods section.

Additional figures S4G, S4H and S5A-A5D were inserted in the supplementary figures file. The “data not shown” phrases were removed from the manuscript.

Additional Editor Comments (if provided):

Dear authors,

The manuscript has been reviewed by three reviewers. This scientific experiment has great potential to inform science. However, there are several areas needed to be clarify before this manuscript can be accepted for publication. The major concerns are related to methodology given there were two studies involved in this manuscript. Clear description of study design, study procedure, inclusion/exclusion criteria, and outcome variables are needed. In addition, in discussion section, the first paragraph is NOT supported by the study results. Addressing these issues can enhance the quality and clarity of the manuscript.

Based on the comments and questions of the reviewers the material and methods section, particularly the description of the study design, was enriched in the missing details. 

We tried to identify the discussion section which is not supported by the presented results. We modified the first paragraph to reflect clearer our interpretation of the results. We hope that the applied changes help understanding our intentions.

Reviewers' comments:

Reviewer's Responses to Questions

Comments to the Author

1. Is the manuscript technically sound, and do the data support the conclusions?

Reviewer #1: Yes

Reviewer #2: No

Reviewer #3: Yes

2. Has the statistical analysis been performed appropriately and rigorously? 

Reviewer #1: Yes

Reviewer #2: Yes

Reviewer #3: Yes

3. Have the authors made all data underlying the findings in their manuscript fully available?

Reviewer #1: Yes

Reviewer #2: Yes

Reviewer #3: Yes

4. Is the manuscript presented in an intelligible fashion and written in standard English?

Reviewer #1: Yes

Reviewer #2: No

Reviewer #3: Yes

5. Review Comments to the Author

Reviewer #1: This study has the potential to contribute significantly to the understanding of visual cues and hunger hormones. The outcome highlighting the highly individualized response to ghrelin has implications for further study and the understanding of hunger behavior in light of the obesity epidemic.

There are no major issues related to the literature review, methodology or presentation of the results. The discussion section could be expanded to include clinical implications of the outcomes and what additional research is needed to forward this science.

Minor issues are listed below.

Title:

This research appears an important contribution to the study of obesity, however, the title and key words do not highlight this point. I would think that the authors would like to increase the visibility among researchers looking at obesity.

The title was modified accordingly.

Introduction:

Page 3, line 67: Suggest elaborating on negative health outcomes related to obesity that will support the significance of your research.

The section concerning negative health outcomes associated with obesity was inserted in the introduction.

Page 4, line 77: Would remove the word hedonistic.

The word “hedonistic” was removed and replaced with “self-indulgent”.

Page 4, lines 82-83: Would take the opportunity to discuss how ghrelin concentration is impaired in obese individuals.

The section concerning ghrelin levels in obese individuals was inserted in the introduction.

Page 4, lines 84-86: Elaborate on the difference between active and inactive forms. What does the difference contribute to this research?

Explanation concerning the roles of acylated and deacylated ghrelin was inserted in the introduction.

Page 5, lines 111-112: What will investigating what form of ghrelin add to your study?

The sentence concerning the unclear role of acylated and deacylated ghrelin in hunger signaling was added to the introduction. 

Page 5, lines 117-123: How were the participants recruited? What were they told about the study? How did you ensure the study participants were not pregnant?

The additional information was introduced in the materials and method section.

Methods:

Page 7. Lines 158-162: Recommend moving Figure 1 discussion to after 2.2 study II.

The figure has been moved as suggested. 

Page 7, lines 165-167: How were the participants recruited? Why did you recruit a difference sample size? How did you ensure the women were not pregnant?

We introduced the missing information in the manuscript. The subjects were recruited via posters and social media posts encouraging participation. 

Concerning the different samples size, we aimed at having at least 20 participants per group and as we expected that several of them would drop out we sought to recruit several candidates more than 20. In study I 23 participants completed the study and we decided that the increased number of samples can only benefit our analysis therefore we used the samples of all participants for the analysis. 

The participants were explicitly asked to declare in a written consent form whether they are pregnant and additionally they were asked to declare the day of their menstrual cycle.

Page 8, lines 184-185: How did you ensure validity and reliability of sampling and analysis procedures?

The sampling procedure was carried through by a professional medical doctor regularly supporting human studies at the University of Vienna. Installation of a venous catheter ensured rapid and reliable collection of specific volumes of blood in a repetitive manner. The measurements of glucose and triglycerides were performed using standard clinical equipment. Hormones levels were assessed using a Milliplex kit and Luminex xMAP reader applying all required standards as well as calibration steps.

Results:

Page 14, lines 335-339: Figure 4 results do not appear to be included in the purpose and aims of the study.

The abstract and introduction have been modified to include the analysis of the visual factors contributing to the response to the sensory stimulation.

Discussion:

Well developed. Would like to have seen a discussion on the clinical implications of the study’s outcomes.

The discussion part has been enriched in section concerning possible applications of the current study and similar studies to follow.

Figure 1: Would include N and NS in labeling “pictures.”

The labels have been introduced in the figure

Figure 2 F&G is confusing. State in key that each color is a participant.

We stated in the figure legend that in the figure 2 F and G each line presents one participant. Introducing a key showing each line with ID of corresponding participant would compromise the esthetic and simplicity of the figure. Additionally, participant’s ID is not an information of importance. On the other hand, putting the information in a written form in the figure (“each like represents one participant”) introduces text in the figure which is difficult to integrate in a visually acceptable way. We could not find a finer way of transmitting the information. If the current version does not satisfy the reviewers we would ask for support and suggestions of changes.

Reviewer #2: Thanks for inviting me to review this article. It sounds an interesting topic but did not address the clinical implication of the study. The design seems not ethical for the participants, drawing eight times of blood sample without telling the purpose of the study. I did not find the statements of the ethical consideration. The writing style is not concise in methodology part. The gap of the literature was not addressed well.

Concerning the ethical aspect, the participants were informed that the study assesses “changes in the blood levels of intestine-derived hormones” along the day (study 1) or “the impact of the nutrients in the milkshake on the hormone levels in the blood” (study 2). Therefore, the participants were aware of the reasoning behind blood sampling and gave informed written consent. The signed consent forms are available on demand. The only part of the study that was obscured was the analysis of the impact of the sensory stimulation in order to avoid the possible impact of conscious processes on the spontaneous reaction to food cues. Thank your comment we become aware of the missing information therefore, we updated the materials and methods section.

Based on all of the questions from reviewers we did our best to fill the information gaps in the description of the study and literature review.

Reviewer #3: An interesting study.

1. The description of the methodology can be further clarified. For instance, is this a two-arm experimental trial? or time-series with phase I and phase II? It is best to define what is the research design of this study.

The manuscript presents two studies with different designs. In each study, the participants are submitted both the control as well as the stimulatory conditions. Therefore, we define the experiments are two independent studies with cross-over design.

2. There are also some unclear areas, such as in the abstract, the authors stated this is a two-day study. However, in the extensive description stated in the method, the days involved in the actual study spanned more than two days in study 1 and study 2. It would be helpful to have schematic diagram to display the start date and end date of the studies, and intervals between and within the studies.

The groups of participants of study 1 as well as study 2 each spent two days taking part in the study. To make the design clearer we modified Figure 1.

3. Another unclear area: Are the participants in both study 1 and 2 the same group of sample, or they are pulled form different sampling groups?

The participants of study 1 and study 2 are two different groups of individuals. The reasoning behind such design is to avoid the impact of predictability of the visual stimulation which could influence the response of the participants.

---

## [Decision Letter · Decision Letter 1]

8 Apr 2020

Visual stimulation with food pictures in the regulation of hunger hormones and nutrient deposition, a potential contributor to the obesity crisis

PONE-D-19-23483R1

Dear Dr. Duszka,

We are pleased to inform you that your manuscript has been judged scientifically suitable for publication and will be formally accepted for publication once it complies with all outstanding technical requirements.

With kind regards,

Zhifeng Gao

Academic Editor

PLOS ONE

Additional Editor Comments (optional):

Reviewers' comments:

Reviewer's Responses to Questions

**Comments to the Author**

1. If the authors have adequately addressed your comments raised in a previous round of review and you feel that this manuscript is now acceptable for publication, you may indicate that here to bypass the “Comments to the Author” section, enter your conflict of interest statement in the “Confidential to Editor” section, and submit your "Accept" recommendation.

Reviewer #1: All comments have been addressed

Reviewer #3: All comments have been addressed

2. Is the manuscript technically sound, and do the data support the conclusions?

Reviewer #1: Yes

Reviewer #3: Yes

3. Has the statistical analysis been performed appropriately and rigorously? 

Reviewer #1: Yes

Reviewer #3: Yes

4. Have the authors made all data underlying the findings in their manuscript fully available?

Reviewer #1: Yes

Reviewer #3: Yes

5. Is the manuscript presented in an intelligible fashion and written in standard English?

Reviewer #1: Yes

Reviewer #3: Yes

6. Review Comments to the Author

Reviewer #1: (No Response)

Reviewer #3: (No Response)

7. PLOS authors have the option to publish the peer review history of their article (what does this mean?). If published, this will include your full peer review and any attached files.

Reviewer #1: No

Reviewer #3: No

---

## [Editor Report · Acceptance letter]

9 Apr 2020

PONE-D-19-23483R1 

Visual stimulation with food pictures in the regulation of hunger hormones and nutrient deposition, a potential contributor to the obesity crisis 

Dear Dr. Duszka:

I am pleased to inform you that your manuscript has been deemed suitable for publication in PLOS ONE. Congratulations! Your manuscript is now with our production department. 

With kind regards,

on behalf of

Dr. Zhifeng Gao 

Academic Editor

PLOS ONE